

# Relationship between locomotive syndrome risk test and height and weight in infancy and toddlerhood: a preliminary study

Shuntaro Tsuji*, Tamaki Hirose*, Yohei Sawaya, Runa Kawano, Takumi Sugiura, Riona Takimoto, Yuto Nishimura and Masafumi Itokazu

Department of Physical Therapy, School of Health Sciences, International University of Health and Welfare, Otawara, Tochigi, Japan
* These authors contributed equally to this work.

Corresponding authors
Tamaki Hirose, n-tamaki@ihwg.jp
Yohei Sawaya, sawaya@ihwg.jp

## ABSTRACT

**Introduction:** The Japanese Medical Science Federation's declaration highlighted that locomotive syndrome (LS) precedes frailty, emphasizing the importance of early intervention beginning in childhood and continuing throughout life. To date, no study has examined the relationship between LS in adults and height and weight during infancy and toddlerhood. As a preliminary investigation, this study aimed to clarify how height and weight during early life stages are related to LS among young adults.

**Methods:** In this study, 62 students enrolled at a health sciences university were administered the locomotive syndrome risk test, and their grip strength was measured. Height and weight at birth, at 1 year, and at 3 years of age were obtained from the Maternal and Child Health Handbook. All participants were born *via* normal delivery. The relationships between the LS risk test results, height, and weight were analyzed using correlation and multivariate analyses.

**Results:** Correlation analysis revealed significant associations between height and weight in infancy and toddlerhood and the following variables: two-step length, two-step test value, 25-question geriatric locomotive function scale score, and grip strength. Multiple regression analysis showed a significant relationship between two-step length and height at birth ($p = 0.002$), and between grip strength and weight at age 3 ($p = 0.021$).

**Conclusions:** A weak-to-moderate relationship was found between LS and height and weight in infancy and toddlerhood. These findings suggest that preventive measures for LS should not be delayed until adulthood; rather, early-life interventions beginning in infancy, toddlerhood, and childhood may be crucial.

## INTRODUCTION

Locomotive syndrome (LS) is a condition in which the mobility functions necessary for standing and walking are impaired owing to locomotor system disorders. The risk of needing long-term care in the future is believed to increase as LS progresses (*Japanese Orthopaedic Association, 2012*; *Nakamura, 2008*; *Ishibashi, 2018*). In 2022, the Japanese Medical Science Federation published the "*Declaration of the Medical Society for Overcoming Frailty and Locomotive Syndrome.*" Since LS develops before frailty, measures from childhood and lifelong approaches are important (*The Japanese Medical Science Federation, 2022*; *Sawaya et al., 2024*). However, there is insufficient evidence to support starting interventions from early childhood, and further accumulation of evidence is needed.

Focusing on the prevalence of LS among young people, we found that the prevalence among university students was 20.8% (*Sawaya et al., 2024*). Previous studies have reported that the prevalence of LS among university students is 14.1–21.7% (*Yasuda, 2021*; *Ma et al., 2023*), and the prevalence of LS among men in their 20s is 19.6% (*Nishimura et al., 2020*). Therefore, it is clear that some young people experience LS. The factors associated with LS in university students include reduced balance, high body fat percentage, musculoskeletal pain, and low skeletal muscle mass (*Yasuda, 2021*; *Ma et al., 2023*; *Sawaya et al., 2024*). However, there are still only a few studies that have explored the prevalence of LS and related factors in young people. In addition, these studies explored factors based on physical function and lifestyle habits in the university phase; there have been no reports analyzing factors related to LS in young adults based on infancy, toddlerhood, or childhood factors.

Many studies have explored the relationship between early childhood indicators such as birth weight and motor function, particularly grip strength, which is a key measure of muscle strength. A systematic review of grip strength reported a positive correlation between birth weight and muscle strength (*Dodds et al., 2012*). Another study found that infants with a low birth weight exhibited poor cardiopulmonary endurance and grip strength (*de Souza et al., 2022*). Furthermore, low birth weight in infancy has been associated with reduced muscle mass and grip strength, whereas weight loss has been linked to an increased risk of sarcopenia in adulthood (*Celik et al., 2024*). Although most studies have focused on birth weight and specific physical functions, no studies have investigated the relationship between LS in adults and height and weight at birth, infancy, or toddlerhood.

Therefore, as a preliminary study, we aimed to clarify the association between LS in young adults and their height and weight in infancy/toddlerhood, as a preliminary study that will contribute to LS measures throughout life, from infancy to toddlerhood. Because LS detects a more subtle decline in motor function than frailty or sarcopenia, early measures of LS are considered the first step toward extending a healthy lifespan.

## MATERIALS AND METHODS

### Study design

This retrospective cohort study was conducted in April 2024. The purpose and methods of the study were explained to the participants verbally and in writing, and written informed consent was obtained before the measurements were conducted. This study was approved by the International University of Health and Welfare Ethics Committee (approval number: 22-Io-34-2) and conducted in compliance with the guidelines of the Declaration of Helsinki.

### Study setting and participants

The participants were first-to fourth-year students enrolled in a health science university. Participants were recruited *via* email and verbal appeals. Of the 115 participants, 22 were excluded because data on height and weight at birth, 1 year, and 3 years of age were missing; 28 were excluded because they did not have a normal delivery; and 3 were excluded because their height and weight at birth, 1 year, and 3 years of age were outliers. As a result, 62 participants (17 males and 45 females; 19.6 ± 1.3 years: mean ± standard deviation) were included in the analysis (Fig. 1). All analyzed participants were born *via* normal delivery.

### Assessment of locomotive syndrome

Locomotive syndrome risk tests, including the stand-up test, two-step test, and 25-question geriatric locomotive function scale (GLFS-25), were administered. The method of administering the locomotive syndrome risk test was performed as described in previous studies. Participants who did not qualify as having LS were classified into the non-LS group, and those who qualified as having locomotive syndrome stage (LS stage) 1, 2, or 3 were classified into the LS group (*Japanese Orthopaedic Association, 2012*; *Sawaya et al., 2024*). The stand-up test used platforms of 40, 30, 20, and 10 cm in height, and the participants were deemed successful if they could stand up on either one or both legs without exerting any momentum and maintain a standing position for 3 s. The order of the platform and eitherone or both legs was determined based on a previous study (*Japanese Orthopaedic Association, 2012*). The stand-up test is scored from 0 to 8, with lower scores indicating more severe symptoms (*Ogata et al., 2015*). For the two-step test, the participants aligned their toes with the starting line and took two steps forward with the maximum stride length, at which point the distance to the position of their toes was measured (*Japanese Orthopaedic Association, 2012*). Measurements were performed twice and the maximum value was used in the analysis. In this study, the actual measured values of the two strides and two-step test values, calculated by dividing the two strides by height, were used for the analysis. The GLFS-25 consists of 25 questions regarding physical and living conditions. The minimum and maximum scores are 0 and 100, respectively, with higher scores indicating more severe symptoms (*Seichi et al., 2012*).

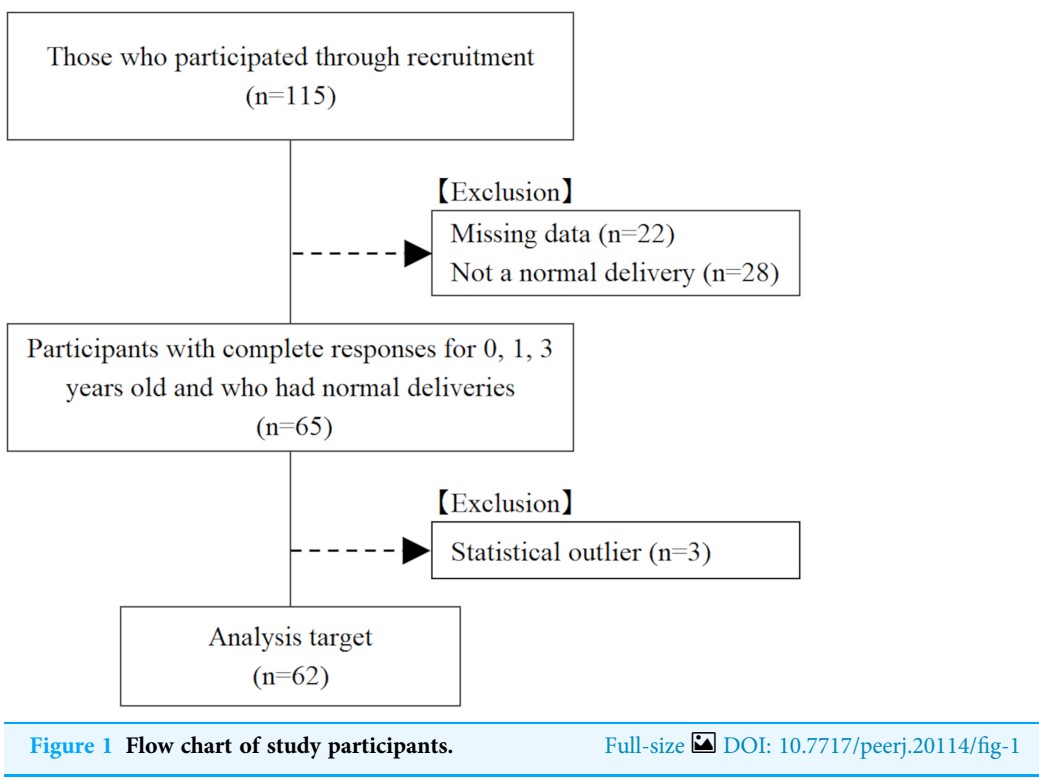

**Figure 1  Flow chart of study participants.**     

In addition, the locomotive syndrome risk tests uses the same clinical cutoff values and evaluation methods across a wide age range, from individuals in their 20s to older adults, and age-specific reference values have also been established (*Japanese Orthopaedic Association, 2012*; *Yamada et al., 2020*). Furthermore, the locomotive syndrome risk tests do not have floor and ceiling effects across various age groups (*Ogata et al., 2015*).

## Assessment of height and weight in infancy and toddlerhood

Using the Maternal and Child Health Handbook, we asked about height and weight at birth, at 1 year and 3 years of age, and the method of delivery (natural delivery, caesarean section, or other). For all items, height was measured in centimeters and weight in grams. In Japan, municipalities issue Maternal and Child Health Handbooks to all mothers; thus, it is possible to obtain accurate physical data on infants and young children as evaluated by medical professionals, such as midwives (*Anabuki et al., 2019*).

## Other assessments

Grip strength was measured twice on each side in the standing position by using a Smedley-type grip strength meter (GRIP-D T. K. K5401; Takei Scientific Instruments Co., Ltd., Tokyo, Japan), and the maximum value was used as the representative value. Based on previous literature, current exercise habits were defined as exercising at least twice a week for at least 30 min each time (*Tabata et al., 2023*). In addition, current height and weight were obtained at the time of measurement, and the body mass index (BMI) was calculated.

## Statistical analyses

Statistical analysis was performed using unpaired t-tests to compare height and weight in infancy and toddlerhood between the groups with and without LS. Pearson's product-moment correlation coefficient and Spearman's rank correlation coefficient were used to analyze the relationship between the locomotive syndrome risk test and grip strength, height, and weight during infancy (at birth, 1 year, and 3 years of age). Third, multivariate analysis was performed. In the multiple regression analysis, in Pattern I, the dependent variable was two-step length, and the independent variables were height and weight during infancy (at birth, 1 year, and 3 years of age). In Pattern II, the dependent variable was grip strength, and the independent variables were height and weight during infancy (at birth, 1 year, and 3 years of age). In multivariate analyses, sex and current exercise habits were included as adjustment variables in Model I and sex, current exercise habits, and current BMI were included in Model II. For missing values of BMI, multiple imputation was performed five times, and the pooled results were calculated using Rubin's rules (*Rubin, 1987*). Independent variables were selected using the stepwise method, taking multicollinearity into account. Statistical analyses were using IBM SPSS Statistics ver29 (IBM Corp., Armonk, NY, USA) with a significance level of 5%. *Post hoc* power analysis was performed using G*Power version 3.1.9.2.

## RESULTS

Of the 62 participants analyzed in this study, 16 (25.8%) were in the LS group and 46 (74.2%) were in the non-LS group. All participants in the LS group had LS stage 1. The other basic attributes of the participants and the measured data are listed in Table 1. Table 2 shows a comparison of height and weight in the infant and toddler groups between those with and without LS. No statistically significant differences were observed in the heights and weights of infants and toddlers between the LS and non-LS groups.

Table 3 shows the correlations between locomotive syndrome risk test results and grip strength, height, and weight during infancy (at birth, 1 year, and 3 years of age). Two-step length and height at birth ($r = 0.430$, $p < 0.001$), weight at birth ($r = 0.438$, $p < 0.001$), height at age 1 ($r = 0.368$, $p = 0.003$), weight at age 1 ($r = 0.356$, $p = 0.005$), and height at age 3 ($r = 0.319$, $p = 0.011$) were significantly correlated. In addition, significant correlations were found between two-step test value and weight at birth ($r = 0.260$, $p = 0.041$), GLFS-25 and height at birth ($\rho = -0.281$, $p = 0.027$), grip strength and height at birth ($r = 0.263$, $p = 0.039$), weight at birth ($r = 0.361$, $p = 0.004$), height at age 1 ($r = 0.308$, $p = 0.015$), and weight at age 1 ($r = 0.418$, $p = 0.001$). The results of the partial correlation analysis controlling for sex are shown in Table S1. Table S2 presents the correlation between height and weight at birth, 1 year, and 3 years of age.

Table 4 shows the results of the multiple regression analysis of the relationships between two-step length/grip strength, height, and weight in infancy/toddlerhood. A significant relationship was found between two-step length and height at birth: Model I ($\beta = 3.725$, 95% confidence interval for $\beta = 1.433$–6.018, standardized coefficient $\beta = 0.308$, $p = 0.002$), Model II ($\beta = 3.699$, 95% confidence interval for $\beta = 1.442$–5.957, standardized coefficient

**Table 1 Basic attributes and measured data.**

| | |
|---|---|
| Age | 19.6 ± 1.3 |
| Sex | |
| Male | 17 (27.4) |
| Female | 45 (72.6) |
| Height (cm) | 161.7 ± 8.4 |
| Weight (kg) | 53.7 ± 9.4 |
| BMI (kg/m$^2$) | 20.4 ± 2.7 |
| Percentage of LS | |
| LS group | 16 (25.8) |
| Non-LS group | 46 (74.2) |
| Stand-up test (point) | 5.5 (5.0–7.0) |
| Two-step length (cm) | 261.2 ± 26.7 |
| Two-step test value (cm/height) | 1.61 ± 0.13 |
| GLFS-25 (point) | 1.0 (0.0–4.3) |
| Grip strength (kg) | 30.6 ± 8.1 |
| Exercise habits | |
| Presence of exercise habits | 47 (75.8) |
| No exercise habits | 15 (24.2) |
| Mother and child health handbook | |
| Height at birth (cm) | 49.0 ± 2.2 |
| Weight at birth (g) | 3,023.5 ± 385.9 |
| Height at age 1 (cm) | 75.6 ± 3.9 |
| Weight at age 1 (g) | 9,297.3 ± 1,212.8 |
| Height at age 3 (cm) | 94.0 ± 4.0 |
| Weight at age 3 (g) | 13,850.0 ± 1,545.8 |

Notes:
BMI, Body Mass Index. GLFS-25:25-question geriatric locomotive function scale. LS, Locomotive syndrome.
Data are presented as mean ± standard deviation, number (%), and median (25–75%).

**Table 2 Comparison of height and weight in infancy/toddlerhood with and without locomotive syndrome.**

| | LS group (n = 16) | Non-LS group (n = 46) | p-value |
|---|---|---|---|
| Height at birth (cm) | 49.5 ± 2.0 | 48.8 ± 2.3 | 0.290 |
| Weight at birth (g) | 3,041.4 ± 247.4 | 3,017.2 ± 425.8 | 0.785 |
| Height at age 1 (cm) | 76.2 ± 3.4 | 75.4 ± 4.1 | 0.497 |
| Weight at age 1 (g) | 9,612.2 ± 1,119.4 | 9,187.8 ± 1,236.4 | 0.231 |
| Height at age 3 (cm) | 94.4 ± 3.3 | 93.8 ± 4.2 | 0.619 |
| Weight at age 3 (g) | 14,247.5 ± 1,697.9 | 13,711.7 ± 1,484.0 | 0.235 |

Note:
The data are presented as mean ± standard deviation.

**Table 3 Correlation between the locomotive syndrome risk test, grip strength, and height/weight at birth, 1 year, and 3 years of age.**

| | Stand-up test | | Two-step length | | Two-step test value | | GLFS-25 | | Grip strength | |
|---|---|---|---|---|---|---|---|---|---|---|
| | ρ | p-value | r | p-value | r | p-value | ρ | p-value | r | p-value |
| Height at birth (cm) | −0.126 | 0.328 | 0.430 | <0.001* | 0.249 | 0.051 | −0.281 | 0.027* | 0.263 | 0.039* |
| Weight at birth (g) | 0.074 | 0.568 | 0.438 | <0.001* | 0.260 | 0.041* | −0.029 | 0.825 | 0.361 | 0.004* |
| Height at age 1 (cm) | −0.230 | 0.072 | 0.368 | 0.003* | 0.126 | 0.331 | −0.089 | 0.493 | 0.308 | 0.015* |
| Weight at age 1 (g) | −0.095 | 0.464 | 0.356 | 0.005* | 0.117 | 0.365 | 0.131 | 0.310 | 0.418 | 0.001* |
| Height at age 3 (cm) | −0.178 | 0.167 | 0.319 | 0.011* | 0.026 | 0.841 | −0.161 | 0.210 | 0.217 | 0.091 |
| Weight at age 3 (g) | −0.173 | 0.178 | 0.176 | 0.171 | −0.032 | 0.804 | −0.089 | 0.493 | 0.195 | 0.129 |

Notes:
The Stand-up Test and GLFS-25 indicate Spearman's rank correlation coefficients.
The two-step test value, two-step test value, and grip strength were calculated using Pearson's correlation coefficients.
* $p < 0.05$.
GLFS-25:25-question geriatric locomotive function scale.

**Table 4 Relationship between two-step length/grip strength and height/weight at birth, 1 year, and 3 years of age analyzed using multiple regression analysis.**

| | Model I | | | | Model II | | | |
|---|---|---|---|---|---|---|---|---|
| | β | 95% CI for β | SC β | p-value | β | 95% CI for β | SC β | p-value |
| Pattern I (Two-step length) | | | | | | | | |
| Height at birth | 3.725 | [1.433–6.018] | 0.308 | 0.002* | 3.699 | [1.442–5.957] | 0.306 | 0.001* |
| Pattern II (Grip strength) | | | | | | | | |
| Weight at age 3 | 0.001 | [0.000–0.001] | 0.152 | 0.021* | 0.001 | [0.000–0.001] | 0.135 | 0.044* |

Notes:
Pattern I: Dependent variable: two-step length. Independent variables were height and weight at birth, 1 year, and 3 years of age (stepwise method).
Pattern II: Dependent variable: grip strength. Independent variables were height and weight at birth, 1 year, and 3 years of age (stepwise method).
Model I: Adjusted Variables: sex and current exercise habits.
Model II: Adjusted Variables: sex, current exercise habits, and current body mass index.
* $p < 0.05$.
CI, confidence interval; SC, standardized coefficient.

β = 0.306, $p = 0.001$). Similarly, a significant relationship was found between grip strength and weight at 3 years of age: Model I (β = 0.001, 95% confidence interval for β = 0.000–0.001, standardized coefficient β = 0.152, $p = 0.021$), Model II (β = 0.001, 95% confidence interval for β = 0.000–0.001, standardized coefficient β = 0.135, $p = 0.044$). The regression model examining the relationship between two-step test values and height and weight during infancy was not statistically significant. The sample size for multivariate analysis was calculated using F-tests, linear multiple regression, and the *post hoc* method. For two-step length, the adjusted $R^2$ was 0.479 in Model I and 0.473 in Model II, with a power of 0.99 in both models. For grip strength, the adjusted $R^2$ was 0.757 in both Model I and Model II, with a power of 1.00 in both models.

## DISCUSSION

This is the first study to investigate the relationship between LS in adults and height and weight in infants and toddlers, using data from the Maternal and Child Health Handbook. The results showed a weak-to-moderate correlation between the locomotive syndrome risk test and height and weight in infancy and toddlerhood (*Akoglu, 2018*), which was also

revealed by multivariate analysis. It should be noted that despite analyzing only individuals born *via* normal delivery, the locomotive syndrome risk test in university students was found to be associated with physical maturity at birth and at 1 year and 3 years of age (height and weight).

The physiological background of this study was based on previous research showing that low birth weight is associated with a reduced number of muscle fibers and that developmental problems related to muscle morphology lead to reduced muscle strength (*Jensen et al., 2007*; *Patel et al., 2012*). The results of this study suggest that this background may be the basis for the association between locomotive syndrome risk test results and height and weight during infancy and toddlerhood.

Premature infants requiring NICU admission, particularly those with extremely low birth weight, are at a higher risk of developmental challenges, including motor or cognitive delays, even as they grow older (*Eichenwald & Stark, 2008*). Consequently, follow-up systems for these individuals are being gradually established. However, the findings of this study suggest that low birth weight *via* normal delivery may be associated with motor function in adulthood. This underscores the need for early interventions for LS rather than delaying measures until adulthood; instead, preventive approaches should be implemented beginning in childhood.

A systematic review of previous research on grip strength and birth weight reported that for every 1 kg increase in birth weight, grip strength increases by 0.86 kg, and that this relationship is maintained throughout life (*Dodds et al., 2012*). A correlation between birth weight/height and weight/height in infancy and toddlerhood was demonstrated in the sub-analyses of this study and previous studies (*Taveras et al., 2009*; *Oshiro et al., 2022*). Fetal growth during pregnancy is positively correlated with body size in early childhood (*Vinther et al., 2023*). Additionally, physical activity levels at age four have been linked to increased physical activity and improved fitness at age nine (*Tigerstrand Grevnerts et al., 2024*), with similar trends observed in follow-up studies of children between seven ages of 7.5 to 12.6 years (*Breau et al., 2022*). Furthermore, low birth weight status has been associated with teachers' evaluations of sports performance during their school years, which has been linked to middle-age physical activity levels in middle age (*Elhakeem et al., 2017*). Moreover, low birth weight is associated with a significantly reduced number of muscle fibers in individuals aged 68–76 years (*Patel et al., 2012*). Taken together, these findings suggest that aspects of growth and physical function may be interconnected throughout the lifespan—from before birth to old age. In recent years in Japan, the life-course approach proposed in the Declaration of the Medical Society for Overcoming Frailty and Locomotive Syndrome has been increasingly emphasized as a framework to be applied seamlessly across all life stages, from childhood to old age (*The Japanese Medical Science Federation, 2022*). This study may serve as a basis for accumulating such evidence.

Both genetic and environmental factors play a role in human birth weight and development. Previous studies have demonstrated that certain genetic factors are associated with birth weight and disease risk (*Mook-Kanamori et al., 2012*; *Liao, Deng & Zhao, 2020*; *Zhang et al., 2023*). In contrast, environmental factors are modifiable and are

known to exert multifaceted effects on both birth weight and subsequent growth. Prior research has shown that maternal nutritional status during pregnancy, educational level, socioeconomic status, and smoking habits are significantly associated with birth weight (*Tyagi et al., 2017*; *Suzuki et al., 2008*). Additionally, growth from infancy through adolescence is influenced by habitual physical activity, which supports the development of the musculoskeletal and nervous systems, and by parenting styles, parent-child relationships, education, and socioeconomic status (*Noble et al., 2015*; *Yoshikawa, Aber & Beardslee, 2012*; *James et al., 2024*; *Mackey et al., 2015*; *Tooley, Bassett & Mackey, 2021*). In this study, based on the premise that both genetic and environmental factors influence human development, we explored the possibility that aspects of musculoskeletal development may serve as targets for early support and preventive intervention, given the observed associations between body size in infancy and toddlerhood and later locomotive function.

The limitations of this study include the lack of data on gestational age, inability to account for other confounding factors such as nutritional status and growth environment during infancy, and small sample size with an imbalance in the participant sex ratios. This study was a preliminary investigation. Future studies with larger sample sizes are needed to conduct sex-specific analyses and examine the association between LS and birth weight categories, following the approach used in prior research on the relationship between birth weight and body composition (*Yasuda, 2024*, *2025*). In future studies, it will also be necessary to perform an *a priori* sample size calculation. Additionally, because height and weight in infancy and toddlerhood were based on information from the Maternal and Child Health Handbook, we analyzed height and weight during infancy (at birth, 1 year of age, and 3 years of age), for which it was easier to obtain unified and accurate information. In recent years, concerns have emerged regarding declining motor function, an increasing number of children being unable to stand on one leg, and rising childhood obesity rates (*Kubo et al., 2022*). Based on these trends, we hypothesized that children with reduced motor function may be at a higher risk of developing LS in the future. To safeguard the health of future generations, the importance of implementing appropriate childhood measures must be reaffirmed, and longitudinal studies are needed to determine whether LS in young adults is associated with adverse outcomes in later life.

## CONCLUSIONS

In conclusion, this study is the first to report the relationship between LS in adults and height and weight in infants and toddlers. The results revealed a weak-to-moderate relationship. When dealing with LS, it is important to take a life course approach that is tailored to each stage from childhood to old age. However, the results of this study suggest that it may be important to take measures throughout one's life, starting from the early stages of infancy and toddlerhood.

## ACKNOWLEDGEMENTS

We would like to express our gratitude to all participants who contributed to this study.

### Funding

This study was funded by the JSPS Grants-in-Aid for Scientific Research (22K17539) and the Foundation for Total Health Promotion. The funders had no role in study design, data collection and analysis, decision to publish, or preparation of the manuscript.

### Grant Disclosures

The following grant information was disclosed by the authors:
JSPS Grants-in-Aid for Scientific Research: 22K17539.

### Competing Interests

The authors declare that they have no competing interests.

### Author Contributions

- Shuntaro Tsuji conceived and designed the experiments, performed the experiments, prepared figures and/or tables, authored or reviewed drafts of the article, and approved the final draft.
- Tamaki Hirose conceived and designed the experiments, analyzed the data, prepared figures and/or tables, authored or reviewed drafts of the article, and approved the final draft.
- Yohei Sawaya conceived and designed the experiments, analyzed the data, prepared figures and/or tables, authored or reviewed drafts of the article, and approved the final draft.
- Runa Kawano performed the experiments, authored or reviewed drafts of the article, and approved the final draft.
- Takumi Sugiura performed the experiments, authored or reviewed drafts of the article, and approved the final draft.
- Riona Takimoto performed the experiments, authored or reviewed drafts of the article, and approved the final draft.
- Yuto Nishimura performed the experiments, authored or reviewed drafts of the article, and approved the final draft.
- Masafumi Itokazu conceived and designed the experiments, authored or reviewed drafts of the article, and approved the final draft.

### Human Ethics

The following information was supplied relating to ethical approvals (*i.e.*, approving body and any reference numbers):

This study was approved by the Ethics Review Committee of the International University of Health and Welfare (approval number: 22-Io-34-2) and conducted in accordance with the Declaration of Helsinki.

## Data Availability

The raw data are available in the Supplemental Files.

## Supplemental Information

Supplemental information for this article can be found online at http://dx.doi.org/10.7717/peerj.20114#supplemental-information.

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
