# Peer review of "Relationship between locomotive syndrome risk test and height and weight in infancy and toddlerhood: a preliminary study"

_PeerJ, doi:10.7717/peerj.20114_

## Round 0.1 · original submission · Major Revisions

· Academic Editor

Major Revisions

Both reviewers have serious concerns about the data collection, reporting, and analysis. Please address these concerns on detail.

Reviewer 1 ·

Basic reporting

While the manuscript explores a novel and important question—the association between early-life anthropometric data and locomotive syndrome (LS) in young adults—there are two critical limitations that severely compromise the scientific validity of the study.

Sample Size and Justification:
The study is framed as an epidemiological investigation, yet the sample size (n = 62) is extremely small. Furthermore, there is no clear justification or rationale for the sample size determination. In the context of observational research aiming to infer associations with potentially wide-ranging public health implications, this lack of statistical power undermines the credibility and generalizability of the findings.

Omission of Current Body Size as a Covariate:
The authors did not include participants’ current anthropometric data (e.g., BMI, height, weight) as covariates in the multivariate analyses. This is a critical oversight, as current body size may confound the observed associations between early-life growth patterns and LS risk test outcomes. Without adjusting for this important factor, the risk of drawing spurious or misleading conclusions is high.

Given these fundamental issues related to both study design and data analysis, I believe that the manuscript cannot be adequately revised in a short time frame. Therefore, I recommend rejection of this submission.

Experimental design

No comment because I mentioned in the "Basic reporting."

Validity of the findings

No comment

·

Basic reporting

Although mentioned in the limitations of the study, environmental factors are an important point. The study does not mention the differences between genetic and environmental factors at all. Please cite previous studies and give the authors' opinions on the extent to which these factors are related.

Experimental design

The subjects are both male and female, but why is there no consideration of gender differences?
Many studies have reported that young adults in Japan tend to be thin, particularly among women, but this trend has not been reported among men. It is believed that there are significant differences between men and women in terms of skeletal muscle mass and other physical characteristics. By evaluating them together, we may be overlooking important gender differences that could be significant.

What is the effect of height and weight during the toddler stage? (Is the size of the fetus during pregnancy related?) This study is understandable if the size at this stage is considered to be adult size, but can it also address LS, which is a concern in old age?

What is the basis for the number of subjects in this study? Please explain the rationale for the sample size. The subjects in this study show significant differences in the ratio of males to females, and the ratio of males to females is also expected to affect the results.

Validity of the findings

There are likely to be large individual differences in height and weight during the toddler stage due to growth. Why is this stage chosen?
Classifying participants as “low birth weight infants,” “normal birth weight infants,” and “high birth weight infants” would make it easier to raise awareness from birth. Why was this classification, which is commonly used in Japan, not used?

Can LS in college age be considered equivalent to LS in old age?
Can height and weight during the toddler stage reflect LS that becomes characteristic in old age?

---

## Round 0.2 · accepted · Accept

· Academic Editor

Accept

Dear Dr. Tsuji and Dr. Sawaya, I congratulate you on the acceptance of this article for publication.

Reviewer 1 ·

Basic reporting

I reviewed the revised manuscript and confirmed that the authors made significant revisions to the review of previous studies, reanalysis, and discussion.

Experimental design

-

Validity of the findings

-